# Divergent synthesis and identification of the cellular targets of deoxyelephantopins

Roman Lagoutte[1,*], Christelle Serba[1,*], Daniel Abegg[1,*], Dominic G. Hoch[1], Alexander Adibekian[1] & Nicolas Winssinger[1]

Herbal extracts containing sesquiterpene lactones have been extensively used in traditional medicine and are known to be rich in α,β-unsaturated functionalities that can covalently engage target proteins. Here we report synthetic methodologies to access analogues of deoxyelephantopin, a sesquiterpene lactone with anticancer properties. Using alkyne-tagged cellular probes and quantitative proteomics analysis, we identified several cellular targets of deoxyelephantopin. We further demonstrate that deoxyelephantopin antagonizes PPARγ activity *in situ* via covalent engagement of a cysteine residue in the zinc-finger motif of this nuclear receptor.

[1] Department of Organic Chemistry, School of Chemistry and Biochemistry, NCCR Chemical Biology, University of Geneva, 30 quai Ernest-Ansermet, Geneva 1211, Switzerland. * These authors contributed equally to this work. Correspondence and requests for materials should be addressed to A.A. (email: Alexander.Adibekian@unige.ch) or to N.W. (email: Nicolas.Winssinger@unige.ch).

The contribution of natural products to our current pharmacopeia and to the identification of important therapeutic targets is well recognized[1,2]. While natural products are the result of a long evolutionary optimization, a number of examples have demonstrated that synthetic modifications beyond the biosynthetically accessible analogues can bring about important pharmacological improvements. Success stories starting with the semisynthetic derivatization of 6-aminopenicillanic acid to enhance β-lactam activity, to the conversion of erythromycin into azithromycin or baccatin III into taxotere have inspired tremendous efforts in natural product synthesis. While a significant portion of bioactive natural products are endowed with reactive functionalities that can engage in covalent interactions with their target, the historic reluctance to develop covalent inhibitor has curtailed interest in this subset of natural products. In a number of cases, these mildly reactive groups are pivotal to the compound's bioactivity. Despite the potential for promiscuous covalent engagement through unspecific reactions, a number of covalent inhibitors display useful selectivity with regards to their targeted protein[3,4] by virtue of the fact that at low inhibitor concentration (μM), the kinetics of unspecific reaction are slow compared with the reaction resulting from a specific inhibitor–target interaction (that is, high effective concentration of reagents). The preponderance of such reactive groups amongst secondary metabolites would suggest that there is an evolutionary advantage to covalent inhibition. For instance, a covalent inhibitor may also be important in displacing an otherwise unfavourable equilibrium with an endogenous ligand[5]. The declining pipeline of traditional small-molecule drugs coupled to the benefit of covalent binding to overcome resistance/selectivity

issues in kinase inhibition, or efficacy in protease inhibition, have led to a recent reconsideration of covalent inhibitors[6–8]. Natural products have played a key role in the drug-discovery process and as probes in chemical biology[9]. This privileged role has inspired many efforts to access natural-product-like libraries by conventional or diversity-oriented synthesis[10–13]. Terpenoids and sesquiterpene lactones certainly stand out for their historical use in medicine and are rich in mildly reactive functionalities that can engage in a covalent interactions[14,15]. Indeed, functional groups such as α-methylene-γ-butyrolactone, α,β-unsaturated reactive ester chain and epoxides are preponderant in this natural product family and are at the source of its rich biological activity[16–18]. For example, both helenalin (Fig. 1) and parthenolide inhibit the NF-κB pathway by covalently inactivating their target[19]. In the case of helenalin, this inhibition has been proposed to result from a covalent crosslinking of two cysteines in p65. Helenalin is broadly used as an anti-inflammatory drug in the form of its natural extract from *Arnica*. Thapsigargin is widely used in cellular biology and covalently inhibits SERCA (Sarco/endoplasmic reticulum $Ca^{2+}$ ATPase)[20]. Arglabin inhibits protein farnesylation without affecting protein geranylation. On the basis of the critical role of farnesylation for H-Ras function (an important oncogenic driver), this compound has been shown to be an effective antitumour agent[21] and a dimethyl amine prodrug of this natural compound is currently used therapeutically. Most recently, α-methylene-γ-butyrolactones also showed promising antibacterial activity by covalently binding to critical transcriptional regulators and inhibiting the virulence of *Staphylococcus aureus*[22].

Extracts of the plant *Elephantopus scaber* have long been used in traditional medicine with deoxyelephantopin being the most

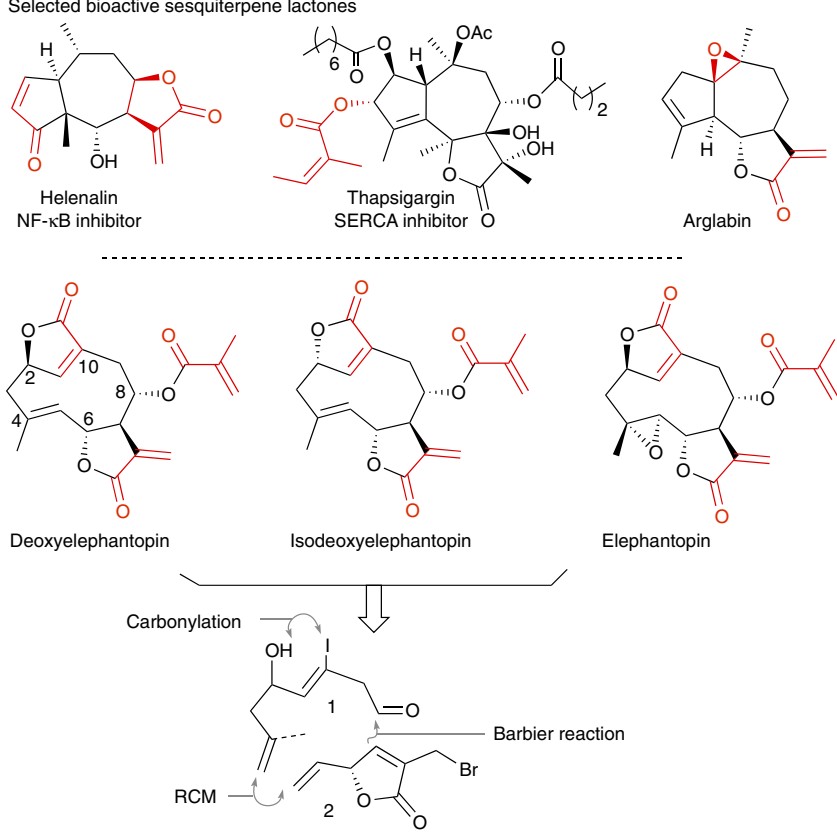

**Figure 1 | Bioactive covalent sesquiterpenes and retrosynthetic analysis of deoxyelephantopins.** Selected examples of bioactive sesquiterpene interacting covalently with their target; structure and retrosynthetic analysis of deoxyelephantopins.

active component[23]. Recently, deoxyelephantopin has been shown to be more effective than paclitaxel in suppressing tumour growth and metastasis in a murine orthotopic breast cancer model[24]. At the cellular level, deoxyelephantopin has been shown to be cytotoxic at doses of 0.5–2 µg ml$^{-1}$ in several human cancer cell lines. While there is evidence that deoxyelephantopin inhibits the NF-κB pathway[24,25], proteomics analysis of up- and downregulated proteome in treated cells suggested it also suppressed proteasome activity[26]. Moreover, SPR experiments suggest that deoxyelephantopin can act as a partial agonist of PPARγ[27], a nuclear receptor that is well known to be involved in pathologies of obesity, diabetes and atherosclerosis and thus represents a major pharmacological target. However, it is not clear whether this natural product can also engage PPARγ directly in cells. While these activities could be rationalized by diverse covalent target engagement, a proteome-wide identification of direct cellular targets of deoxyelephantopin has not been performed to date. The impressive *in vitro* and *in vivo* activities reported for deoxyelephantopin coupled to its historical use as a traditional remedy demands a better understanding of its reactivity profile in a cellular setting and covalent protein target(s). Perhaps, due to its abundance from natural extracts, there is no total synthesis of deoxyelephantopin reported to date nor structure-activity relationship for this promising therapeutic. Here we report synthetic methodologies to access analogues of deoxyelephantopins, including alkyne-tagged cellular probes for quantitative proteomics analysis. We identified several cellular targets of deoxyelephantopin and demonstrated that deoxyelephantopin antagonizes PPARγ activity through covalent engagement.

## Results

**Synthesis of deoxyelephantopins and analogues**. The synthesis of deoxyelephantopin analogues was envisioned to proceed as shown in Fig. 1 making use of a Barbier reaction (conditions are known for *syn*[28] or *anti*[29] addition products) and a ring-closing metathesis (RCM) to join two readily available fragments (**1** and **2**). Mindful that there are few precedents for the RCM of strained 10-membered rings[30] with a triply substituted alkene, we reasoned that the strategy might provide a rapid entry into the hitherto unknown nordeoxyelephantopin and its analogues.

The synthesis commenced with the alkylation of butenal **3** with lithiated alkyne **4** affording the alcohol required in the subsequent hydroxyl-directed alkyne conversion to vinyl iodide **5** (Fig. 2). Nickel-catalysed carbonylation[31] yielded the first key intermediate **6**. The second fragment **2** was assembled from the pentadienol **7**. Acylation with acryloyl chloride followed by a Baylis–Hillman reaction with formaldehyde provided the cyclization substrate **8** that was engaged with Grubbs II followed by a conversion of the allylic alcohol to the bromide under Appel conditions to yield **2** as a racemate. While the prochiral substrate **8** might be desymmetrized via enantioselective RCM with the newly developed ruthenium-based catalysts[32–34], we anticipated to separate diastereomers arising from this racemic fragment at a latter stage. Conversion of the dimethyl acetal **6** to the corresponding aldehyde using hydrated iron trichloride followed by Barbier coupling with allylic bromide **2** afforded the addition product **9** with >9:1 relative stereochemistry for the two newly established stereocenters at C-7 and C-8 and a 1:1 diastereomeric mixture based on the relative stereochemistry of both fragments. With the intermediate in hand, we proceeded to experiment with the RCM. All attempts to carry out a RCM directly on **9** failed using various catalysts. However, introduction

of the methacrylate side chain (**10**) provided an intermediate that did afford nordeoxyelephantopin in the RCM with the desired *E*-alkene geometry. Interestingly, the RCM proceeded only under the action of first generation catalyst (Grubbs I). Furthermore, only the diastereomer corresponding to deoxyelephantopin relative stereochemistry afforded the cyclization product, the C-2 epimer failed to give cyclization. A high correlation between nordeoxyelephantopin and deoxyelephantopin NMR coupling constants suggests that the two products have very similar dihedral angles along the 10-membered ring and hence, a comparable conformation. Furthermore, comparison of NOESY spectra showed similar interactions between the protons on either face of the 10-membered ring for both compounds (Supplementary Figs 1 and 2). With these ring-closing conditions in hand, the same strategy was pursued to access analogues wherein one of the exocyclic conjugate acceptors was reduced (**11** and **12** respectively). While the same reaction starting with **3b** afforded the cyclization precursor (not shown), no cyclization product was observed under a variety of metathesis conditions.

To control the stereochemistry at C-2, we first explored recent chemistry for enantioselective alkyne addition[35], however, substrate **3a** proved problematic based on its propensity to isomerize under Lewis acidic conditions. As an alternative, substrate **13** was converted to the furan **14**, which underwent a palladium-catalysed decarboxylative asymmetric allylic alkylation (DAAA)[36,37] using Trost's ligand to afford **6** in either stereochemistry with 92:8 er.[38] Enantiomerically enriched (*R*)-**6** was used to obtain nordeoxyelephantopin in four steps while (*S*)-**6** (obtained with the (*R*,*R*)-DACH-phenyl catalyst—not shown) afforded the ent-nordeoxyelephantopin.

We reasoned that performing the cyclization before the carbonylation used to form the endocyclic lactone might relax the conformational bias and change the outcome of the cyclization. To this end, substrate **15** (Fig. 3) was prepared according to the same methodology as shown in Fig. 1. Concomitant silylation of the C2 hydroxyl and deprotection of the THP under the action of TESOTf[39] followed by Dess–Martin periodinane oxidation afforded the desired aldehyde that was engaged in the Barbier coupling to yield **16**. Introduction of the methacrylate side chain, or an acetate, followed by exposure to Grubbs II yielded the 10-member cyclized product however, exclusively as the *Z*-alkene (**17a** and **17b**, respectively). As for the previous examples, only one of the C-2 epimers underwent RCM. Silyl deprotection under acidic conditions and nickel-mediated carbonylation afforded **19a** and **b**, the *Z*-analogue of norisodeoxyelephantopin and its C-8 acetate analogue. Structural analysis revealed a close proximity of the allylic C-6 hydrogen to the p-orbital of C-1 conjugated double bond. Exposure of this compound to 254 nm light afforded **20** quantitatively (for related photochemical transformation in sesquiterpene lactones; see ref 40). This transformation presumably proceeds through an excitation of the C1=C10 double bond resulting in C-6 hydrogen abstraction and C-6● C-10● bond formation. To the best of our knowledge, this ring system is unprecedented. Alternatively, intermediate **17a** was engaged in a palladium-catalysed coupling that resulted, after acidic deprotection, in an aromatization affording **18**.

Preliminary experiments on the reactivity of the three conjugate systems of deoxyelephantopin revealed that the γ-butyrolactone was most reactive (reaction with 5 equivalent of glutathione led to a single addition product onto the γ-butyrolactone, see Supplementary Figs 3–5 and Supplementary Methods for conditions). The endocyclic conjugate system proved unreactive. On the basis of this observation, we used the Barbier methodologies to prepare a set of minimal analogues tagged with an alkyne (**23a-c** and **24a–c**, Fig. 4). In addition, pentenal **26** and

**Figure 2 | Synthesis of nordeoxyelephantopin and related analogues.**

2-vinylbenzaldehyde **28** were used to make simplified analogues of the deoxyelephantopin (**26** and **28**, respectively).

**Cytotoxicity of deoxyelephantopins and analogues**. We began our biological investigations by assaying the cytotoxicity of natural deoxyelephantopin (DEP) and its unnatural analogues in four different cancer cell lines (Fig. 5a, Supplementary Figs 6–8). As expected, deoxyelephantopin proved potently toxic (<30% cellular viability at 1 μM concentration). The cytotoxic effect was equally strong when MCF7 cells were treated with nordeoxyelephantopin

or the open ring derivative **10** (Supplementary Fig. 8). We confirmed that the cell death is caused by caspase-mediated apoptosis by treating MCF7 cells with 20 μM DEP and imaging caspase activation and apoptosis using the fluorescent probe FITC-VAD-FMK or Annexin-V-FLUOS staining (Fig. 5b). In contrast, propidium iodide staining did not indicate any significant necrotic death in treated cells (Supplementary Fig. 9).

**Covalent interactome of deoxyelephantopin**. In our efforts to determine the direct molecular targets of deoxyelephantopin we

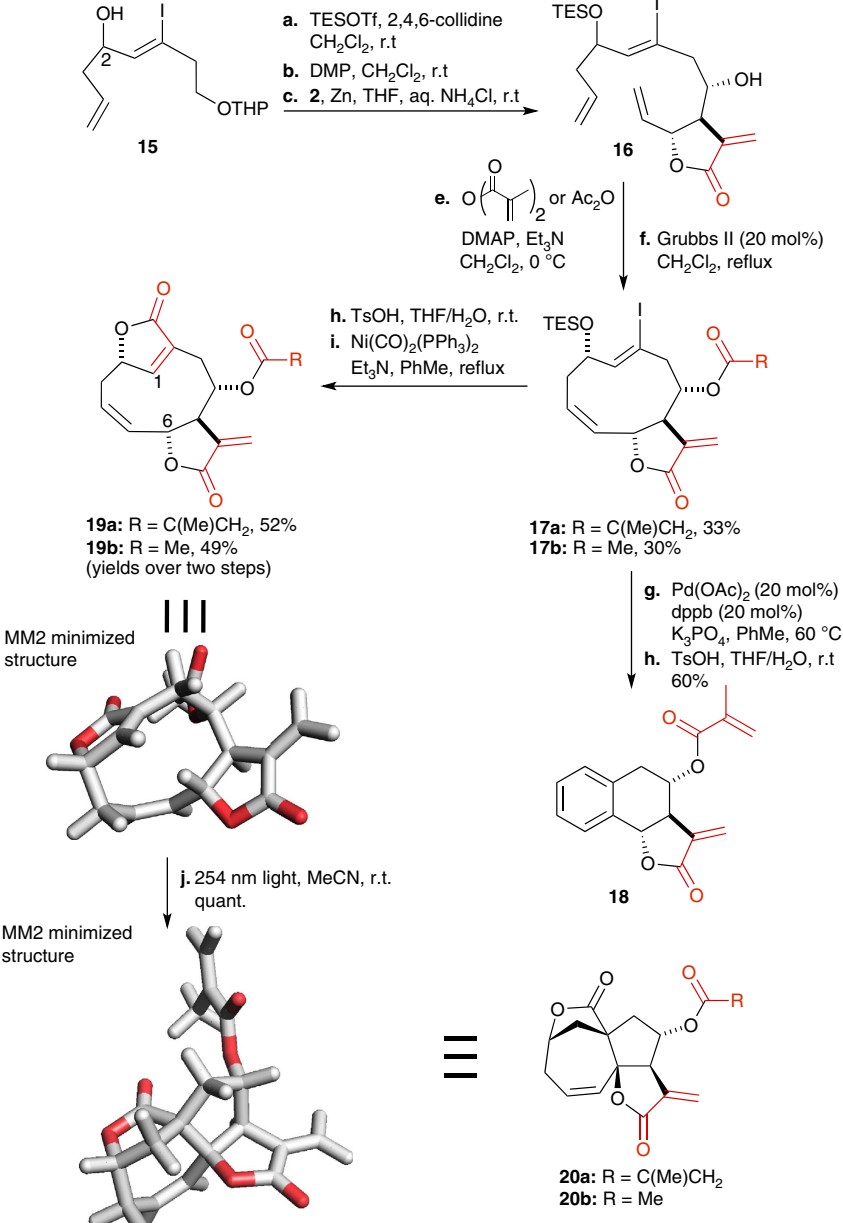

**Figure 3 | Synthesis of deoxyelephantopin-related analogues 18–20.**

first performed a gel-based competitive *in situ* proteomic profiling assay[41] (Fig. 6a, Supplementary Fig. 11) using the γ-butyrolactone probe **24c** that showed the highest cytotoxicity among the entire series of alkyne-tagged analogues **24** in all four tested cell lines (Fig. 5a). While the greater conformational flexibility of probe **24c** could potentially lead to more promiscuous target engagement relatively to deoxyelephantopin, it was favoured based on the fact that an alkyne moiety appended to the rigid scaffold of deoxyelephantopin may hinder some of its interactions. MCF7 cells were pretreated with various concentrations of deoxyelephantopin for 4 h, harvested, lysed and the lysates were treated with the fluorogenic probe **24c**-**Cy3** (10 μM; Supplementary Fig. 10). The gel profile showed multiple labelled bands, but only some of them were selectively competed by twofold excess of the natural product. Furthermore, cells were pretreated with 20 μM concentration of deoxyelephantopin or **24c** and the proteomic profiles were compared (Fig. 6b, Supplementary Fig. 11). Multiple bands were detected that were

equally competed by both DEP and **24c**, thus again confirming that **24c** can be used as an acceptable clickable analogue of deoxyelephantopin in pulldown assays. Having established the optimal labelling conditions by SDS–PAGE, we next carried out a mass spectrometry-based competitive profiling assay[42] coupled with the method of stable isotope labelling of amino acids in culture (SILAC)[43]. In total, 1,522 proteins were enriched from MCF7 cells by 10 μM probe **24c**, but labelling of only 11 proteins was > 70% competed by 20 μM deoxyelephantopin (SILAC ratio < 0.30; Supplementary Fig. 12; Supplementary Data 1). Intriguingly, none of the 11 proteins were previously described as targets of deoxyelephantopin or related natural products. Moreover, we performed a targeted competitive profiling assay following an *in situ* 4 h pretreatment of MCF7 cells with DEP and confirmed that all 11 identified targets are also engaged and bound by deoxyelephantopin directly in living cells (Fig. 6c, Supplementary Data 2). It should be noted that some of the identified targets, including CTTN[44], CSTB[45] and CBS[46], are

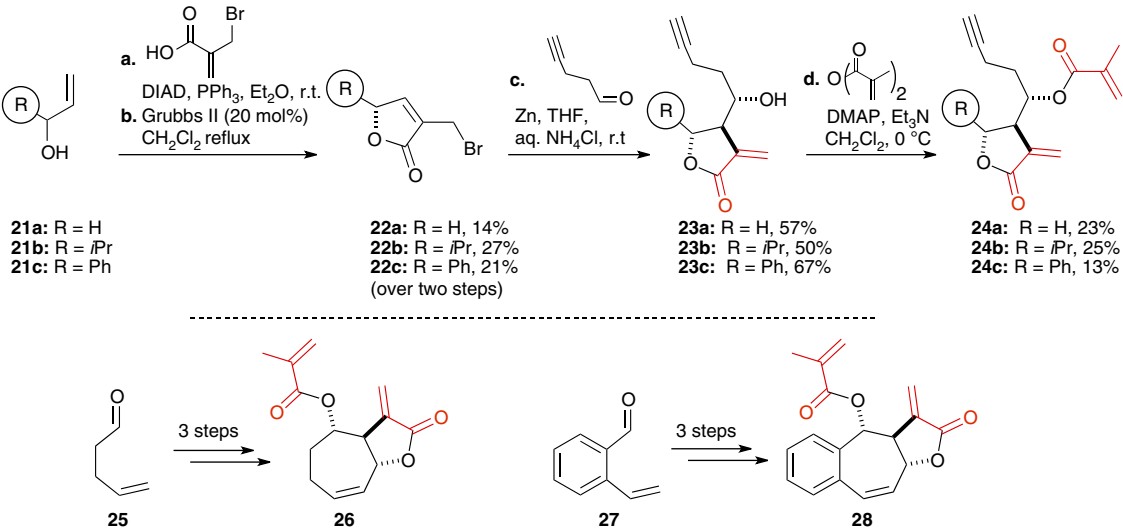

**Figure 4 | Synthesis of deoxyelephantopin-related probes 23 and 24 and simplified analogues 26 and 28.**

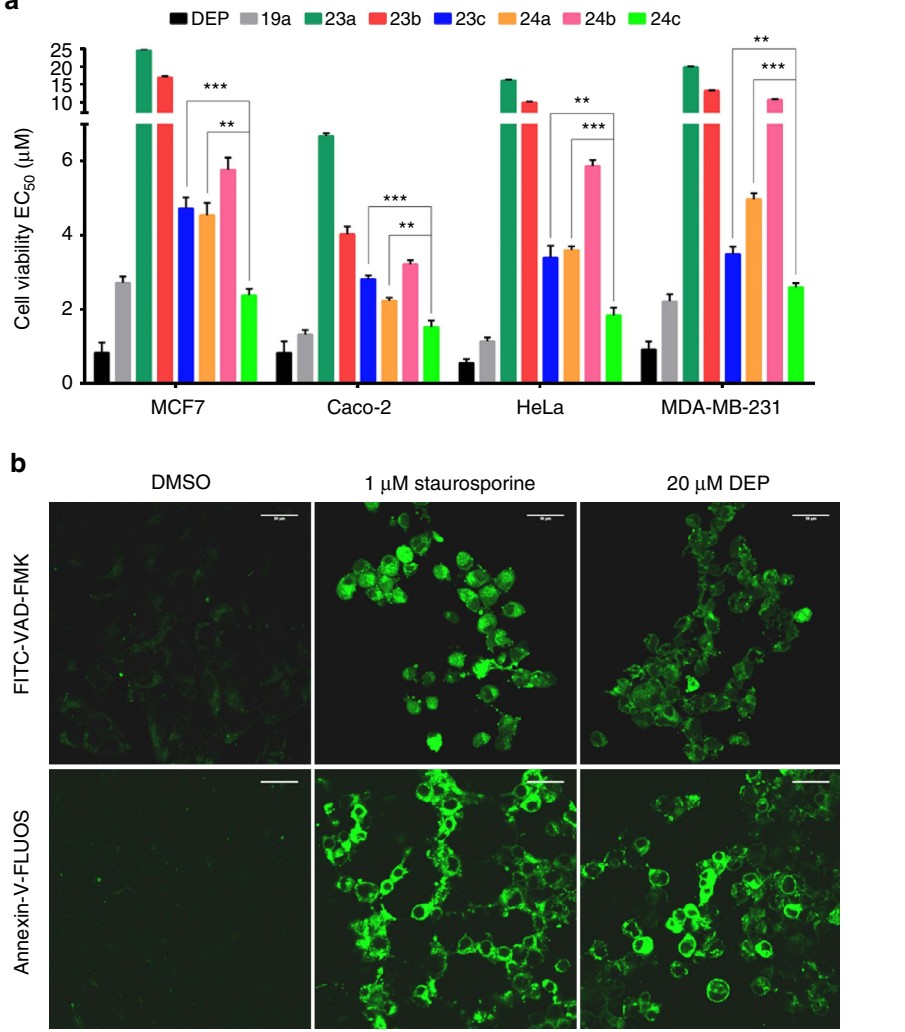

**Figure 5 | Assessing the cytotoxicity of deoxyelephantopin (DEP) and analogues.** (**a**) Cell viability measurements in four different cancer cell lines. Cells were treated with compounds for 3 days, the nuclei were stained with DAPI and counted (EC$_{50}$ values ± s.d. shown; $n = 3$). **$P < 0.005$ and ***$P < 0.0005$ by two-sided Student's $t$-test. (**b**) Detection of caspase activation and apoptosis in MCF7 cells using FITC-VAD-FMK fluorescent probe (top) or Annexin V staining (bottom). The cells were treated with 1 μM staurosporine (positive control) or 20 μM deoxyelephantopin for 8 h. Confocal microscopy images are shown. EC$_{50}$, effector concentration for half-maximum response. Scale bars, 30 μm.

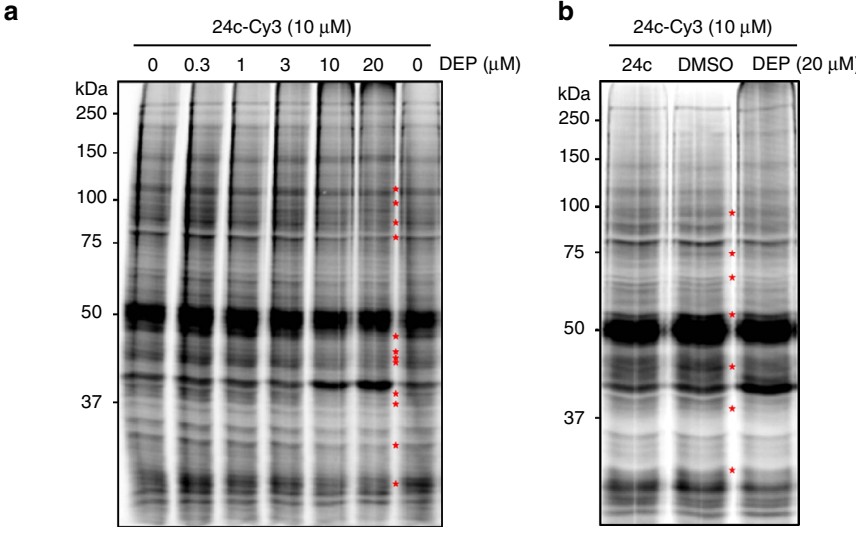

**Figure 6 | Proteome-wide identification of deoxyelephantopin targets.** (**a**) Gel-based competitive *in situ* profiling of deoxyelephantopin (DEP) targets. MCF7 cells were treated with various concentrations of DEP for 4 h, lysed and treated with 10 μM **24c-Cy3**. Asterisks indicate targets competed by DEP. (**b**) Comparison of *in situ* proteomic profiles of DEP versus derivative **24c** after treatment of MCF7 cells for 4 h. (**c**) *In situ* proteomic targets of DEP (>70% competition) in MCF7 cells. Quantification was performed using label-free quantification (LFQ) method ($n = 4$).

| Protein name | Gene name | Mol. weight [kDa] | LFQ ratio DEP/DMSO | s.d. |
|---|---|---|---|---|
| Cystathionine beta-synthase | *CBS* | 60.59 | 0.00 | 0.000 |
| Protein HGH1 homologue | *HGH1* | 42.13 | 0.00 | 0.000 |
| Zinc finger protein 346 | *ZNF346* | 29.27 | 0.00 | 0.000 |
| Glyoxylate reductase/hydroxypyruvate reductase | *GRHPR* | 35.67 | 0.00 | 0.000 |
| Thioredoxin domain-containing protein 12 | *TXNDC12* | 19.21 | 0.04 | 0.001 |
| Heme oxygenase 2 | *HMOX2* | 41.67 | 0.14 | 0.002 |
| Protein transport protein Sec24C | *SEC24C* | 118.32 | 0.18 | 0.031 |
| FLYWCH family member 2 | *FLYWCH2* | 14.56 | 0.19 | 0.010 |
| Src substrate cortactin | *CTTN* | 61.59 | 0.22 | 0.067 |
| Nucleoporin p54 | *NUP54* | 50.78 | 0.24 | 0.034 |
| Cystatin-B | *CSTB* | 11.14 | 0.28 | 0.012 |

known to cause cell death on depletion and thus potentially explain the cytotoxic activity of deoxyelephantopins. Mindful that another sesquiterpene lactone (ainsliadimer A) bearing an α-exo-methylene-γ-butyrolactone has been shown to covalently inhibit IKKα/β (ref. 47), we tested the probe **24c** against recombinant IKKβ but found no evidence of covalent interaction supporting the orthogonal target selectivity of these natural products.

On the basis of the reported interaction between deoxyelephan-topin and PPARγ and because MCF7 cells are known to express very low levels of this protein, we investigated in a separate experiment whether deoxyelephantopin indeed covalently binds PPARγ. The recombinant transcription factor was spiked into MCF7 lysates to 30 nM final concentration and successfully enriched by the methacrylate ester probe **24c**, as evidenced from LC–MS/MS-based label-free quantification (LFQ)[48]. Moreover, PPARγ was also successfully enriched with probe **23c** yielding comparable LFQ value, indicating that the γ-butyrolactone group is involved in the covalent bond formation with the target protein (Supplementary Fig. 13; Supplementary Data 3). To investigate

whether DEP is also capable of binding to endogenous PPARγ directly in living cells, we performed a targeted competitive proteomics experiment in Caco-2, a colon cancer cell line known to express larger amounts of PPARγ[49]. The cells were treated *in situ* with dimethyl sulfoxide (DMSO) or 20 μM deoxyelephantopin, lysed and treated with **24c**. Gratifyingly, endogenous PPARγ was successfully enriched and quantified from the DMSO-treated Caco-2 cells, whereas DEP pretreatment efficiently competed PPARγ enrichment (>95%; Fig. 7a). In yet another control experiment, we expressed hPPARγ in HeLa cells and successfully conducted a gel-based competition experiment once again using DEP as an *in situ* competitor and **24c-Cy3** as reporter probe (Supplementary Fig. 14). Encouraged by these results, we sought to investigate the exact mode of DEP action on PPARγ receptor.

**Deoxyelephantopin antagonizes PPARγ.** Caco-2 cells were treated with DMSO, PPARγ agonist rosiglitazone, antagonist T0070907 or DEP for 24 h, the cells were collected and the

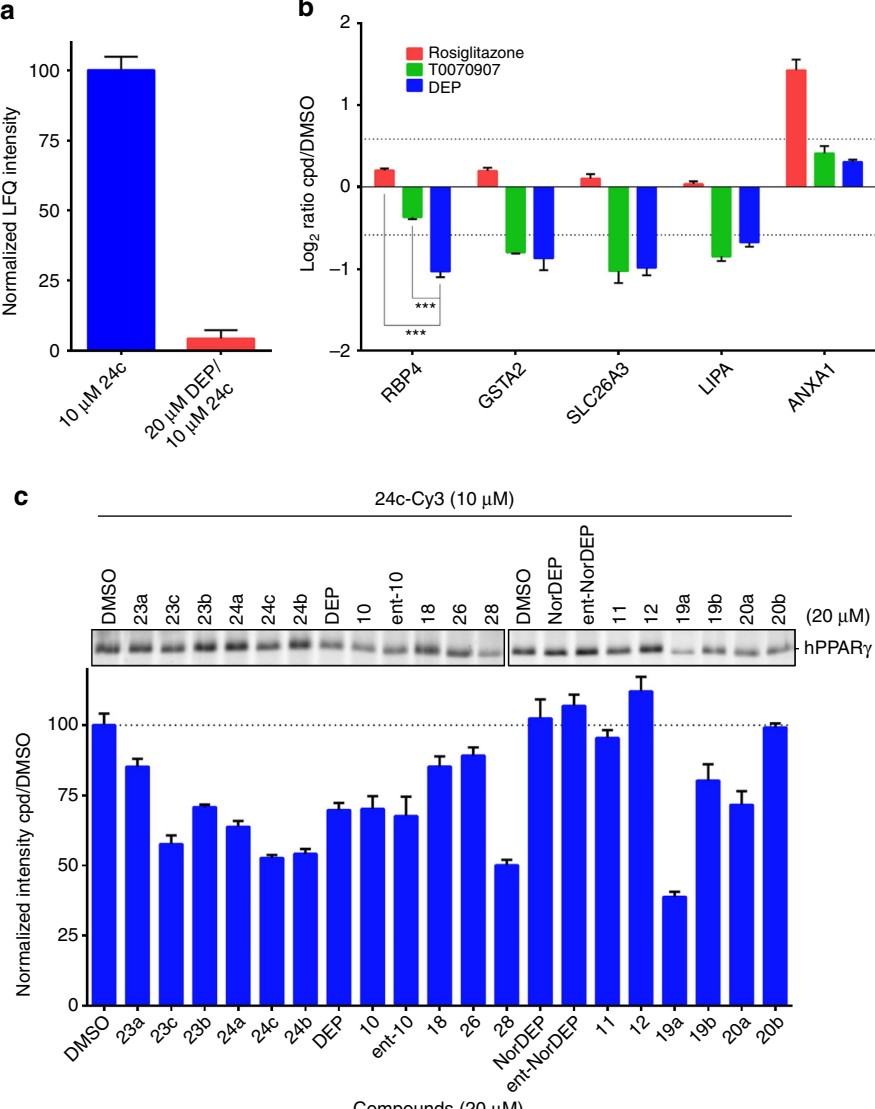

**Figure 7 | Identification of PPARγ as target of deoxyelephantopin and screening of DEP derivatives as PPARγ antagonists. (a)** Quantification of endogenous PPARγ enriched from Caco-2 cells using 10 μM **24c** following *in situ* treatment with DEP (20 μM) or DMSO (normalized LFQ values ± s.d. are shown; n = 6). **(b)** Changes in expression levels of five PPARγ-dependent proteins following 24 h treatment of Caco-2 cells with DMSO, rosiglitazone (10 μM), T0070907 (50 μM) or DEP (20 μM; LFQ ratios ± s.d. are shown; n = 3). **P < 0.005 and ***P < 0.0005 by two-sided Student's t-test. **(c)** Gel-based competitive screening of DEP and analogues as PPARγ antagonists against recombinant human enzyme (top). hPPARγ was treated with each compound (20 μM) for 1 h followed by treatment with **24c-Cy3** (10 μM) as probe. Quantification of fluorescence intensity in the competitive screening for PPARγ antagonists (bottom; values ± s.d. are shown; n = 3).

proteomes were comparatively analysed by LC–MS/MS. Using LFQ as quantification method, we identified 25 upregulated and 18 downregulated proteins following treatment with both DEP and T0070907 and these proteins were either not, or differently affected by rosiglitazone treatment (Supplementary Data 4), indicating that DEP indeed acts in cells as an antagonist of PPARγ. Furthermore, the list of 43 dysregulated proteins was analysed and classified using the GOrilla pathway analysis tool (Supplementary Fig. 15). Further analysis revealed five previously reported PPARγ-dependent proteins RBP4, GSTA2, SLC26A3, LIPA and ANXA1 among all dysregulated targets (Fig. 7b). Most significantly, RBP4, a well-studied adipokine and contributor to insulin resistance in obesity and type 2 diabetes[50] and thus a major therapeutic target, was more strongly downregulated following treatment with DEP than T0070907, suggesting that DEP or a derivative may indeed find clinical use as means to

reduce insulin resistance in various metabolic diseases. Interestingly, T0070907 was also reported to suppress breast cancer cell proliferation and motility in a PPARγ-dependent manner[51], meaning that at least some part of the cytotoxic activity of deoxyelephantopin may indeed originate from antagonizing this receptor.

**Ranking of covalent PPARγ binders.** Wondering whether we would be able to identify more potent PPARγ binders, we also screened other synthetic analogues of deoxyelephantopin in a gel-based competitive format using **24c-Cy3** (Fig. 7c, Supplementary Fig. 16). Briefly, recombinant human PPARγ was pretreated with compounds at 20 μM concentration followed by the probe **24c-Cy3**. Analysis of labelling revealed that the Z-analogue of nordeoxyelephantopin (**19a**) is the most effective

covalent binder to PPARγ followed by the tricyclic analogue **28**. Assuming covalent bond formation with PPARγ, we applied the method of Kitz and Wilson[52] to determine the kinetic values $K_i$ and $k_{inact}$ for deoxyelephantopin and compound **19a** (Supplementary Fig. 17). We observed clear time-dependent shift in half-maximal inhibitory concentration (IC$_{50}$) values characteristic for an irreversible mode of binding. Calculated $k_{inact}/K_i$ values of $680\,M^{-1}\,min^{-1}$ for deoxyelephantopin and $1,241\,M^{-1}\,min^{-1}$ for **19a** confirmed that the latter compound is indeed a more potent PPARγ binder.

**Deoxyelephantopin and related analogues react with PPARγ's zinc finger.** Finally, we sought to understand the molecular basis for PPARγ binding by the synthetic analogues of deoxyelephantopin through identification of the precise covalently modified amino acid site. There is literature precedent for oxidized fatty acids containing an α,β-unsaturated ketone to form a covalent bond with a cysteine in the ligand-binding site of PPARγ[53]. More recently, synthetic ligands binding reversibly to a distinct site were identified[54]. Pretreatment of PPARγ with the general cysteine-reactive probe iodoacetamide (IAA) completely abolished gel-based fluorescence labelling with **24c-Cy3**, implying that probe **24c** and deoxyelephantopin covalently engage a cysteine residue (Supplementary Fig. 18). Bearing in mind that the structurally simplified, and thus less prone to MS/MS fragmentation, probe **23c** binds PPARγ as efficiently as **24c** (Supplementary Fig. 13), we treated the recombinant nuclear receptor with the former compound, digested the protein and analysed the resulting peptides by LC–MS/MS. Peptide mass adduct search and detailed interrogation of MS$^2$ spectra resulted in unambiguous identification of Cys190 as the protein site modified by **23c** (Supplementary Fig. 19). Moreover, we also performed an IAA-competitive labelling experiment, where iodoacetamide labelling of cysteines on PPARγ was competed through pretreatment with compounds **23c**, **24c** and DEP. MS$^2$ spectra inspection revealed efficient (>75%) competition of labelling of the Cys190-containing tryptic peptide with all three compounds, thus additionally confirming the identified binding site (Supplementary Data 5). Interestingly, analysis of the high-resolution X-ray structure of PPARγ (PDB: 3DZU) revealed that this cysteine is coordinated to a $Zn^{2+}$ ion in a zinc-finger motif as part of the DNA binding domain of PPARγ (Supplementary Fig. 20). While zinc fingers respond to oxidative stress through reaction at cysteines[55,56] and have been targeted by

small molecules forming disulfide bridges[57], to the best of our knowledge, deoxyelephantopin is the first example of a small molecule that engages a zinc finger through a Michael addition. The success of Michael acceptors as a cysteine trap in current clinical development of covalent inhibitors points to the privileged reactivity profile of this moiety. Having identified the exact binding site, we then performed covalent docking of **19a** using the protein–ligand docking program GOLD. The predicted optimal binding mode shows key interactions between the methacrylate carbonyl group of **19a** and Asp174 as well as the γ-butyrolactone group and Cys176 (Fig. 8a, Supplementary Fig. 20). Following the docking result, we mutated these two residues, transiently expressed the wild-type and the two PPARγ mutant forms in 293T cells and performed gel-based competitive experiments to determine relative binding affinities for **19a**. We observed a shift in the IC$_{50}$ value from 19 μM (wt PPARG) to 33 μM (C176A) and 58 μM (D174A; Fig. 8b, Supplementary Fig. 21) that further supports the suggested binding mode of **19a**. Finally, it is interesting to note that SILAC experiment led to the identification of ZNF346, another zinc finger motif-containing protein.

In conclusion, the procedures developed in the context of the synthesis of deoxyelephantopin analogues provide a rapid access to diverse ring systems embedding an α-methylene-γ-butyrolactone, an important moiety in sesquiterpene lactones. PPARγ-targeting small molecules such as rosiglitazone and other thiazolidinediones[58] are clinically approved drugs against diabetes. The present discovery that deoxyelephantopin and related synthetic analogues react with the zinc-bound Cys190 in a zinc-finger motif of PPARγ offers a novel pharmacological mechanism for modulating PPARγ activity and may serve as a blueprint for the development of a new generation of potent antagonists of PPARγ (refs 59,60) or other transcription factors with zinc-finger motif. Finally, following the identification of cancer-related proteins CTTN, CSTB and CBS as novel proteomic targets of deoxyelephantopin, it is also tempting to speculate about potential biomedical application of synthetic analogues of this intriguing natural product as novel anticancer therapeutics.

## Methods

**Gel-based screening for PPARγ binders using probe 24c-Cy3.** Recombinant human PPARγ (Cayman Chemical, 50 ng in 12.5 μl PBS) was treated with 20 μM of each compound (0.5 μl of 25 × stock in DMSO), 10 mM iodoacetamide or DMSO for one hour. Samples were then treated with 10 μM **24c-Cy3** (0.5 μl of 25 × stock in DMSO) for 1 h at room temperature in the dark. SDS–PAGE reducing loading buffer (4 ×) was added and proteins were separated using a 10% SDS–PAGE gel.

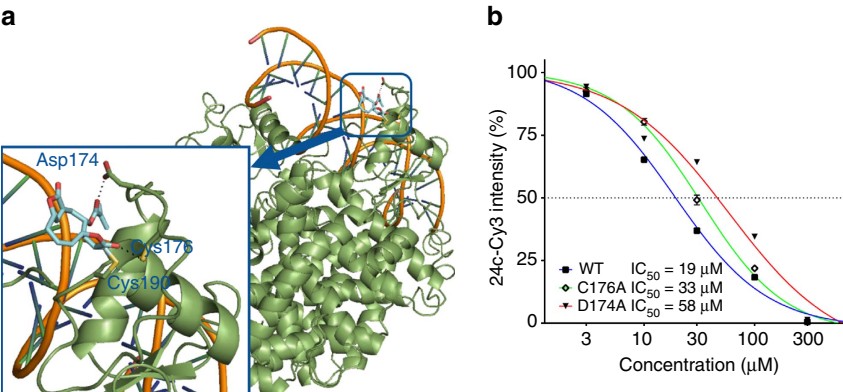

**Figure 8 | Proposed model for 19a binding to PPARγ.** (**a**) Docking prediction of **19a** covalently bound to Cys190 of human PPARγ. The model was created using the protein–ligand docking program GOLD. (**b**) Measurement of **19a** binding to WT, C176A and D174A PPARγ using gel-based competitive assay. The proteins were transiently expressed in 293T cells and the cellular lysates were treated with different concentrations of **19a** for 1h followed by **24c-Cy3** (10 μM). Dose–response curves are shown.

Gels were visualized at 625 nm using a Hitachi FMBIO II Multi-View fluorescence scanner, then stained using silver staining. Images were quantified with ImageJ.

**Detection of a covalent adduct on human PPARγ.** Recombinant human PPARγ (500 ng in 9 μl PBS) was treated with 100 μM of **23c** (1 μl of 10 × stock in DMSO) or DMSO for 1 h. Samples were denatured with 6 M urea in 50 mM $NH_4HCO_3$, reduced with 10 mM TCEP for 30 min and alkylated with 25 mM iodoacetamide for 30 min in the dark. Samples were diluted to 2 M urea with 50 mM $NH_4HCO_3$, and digested with trypsin (0.25 μl of 0.05 μg μl$^{-1}$) in the presence of 1 mM $CaCl_2$ for 12 h at 37 °C. Samples were acidified to a final concentration of 5% acetic acid, desalted over a self-packed C18 spin column and dried. Samples were resuspended in 0.1% FA in water and analysed by LC–MS/MS. The theoretical mass of the **23c** adduct was calculated and LC–MS/MS was run with an inclusion list containing $m/z$ and charge of the expected peptides. MS data were analysed in MaxQuant with **23c** ( + 256.1099 Da) and carbamidomethylation as variable modifications on cysteine. Only modified peptides with PEP value ≤1% were considered.

**Data availability.** The data supporting the findings of this study are available within the article and its Supplementary Information (Supplementary Figs 1–98; Experimental procedures for cellular and biochemical experiments; NMR comparison of deoxyelephantopin and nordeoxyelephantopin; NMR spectra and chiral GC chromatograms); Supplementary Data 1 (Supplementary Tables 1–5 of the proteomic data) and from the corresponding author on request.

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

## Acknowledgements

We thank the SNSF and the NCCR Chemical Biology for financial support.

## Author contributions

N.W. and A.A. conceived the study and designed the experiments. R.L. and C.S. synthesized the compounds. D.A. and D.G.H. performed the biological studies. N.W. and A.A. wrote the manuscript, all authors reviewed and edited the manuscript.

## Additional information

**Competing financial interests:** The authors declare no competing financial interests.

