## [Peer review file · Nature Communications]

Reviewer #1 (Remarks to the Author):

In the manuscript entitled 'Divergent synthesis of deoxyelephantopins and investigations into its covalent interactome' by Winssinger et al. authors describe synthesis and biological activity of (+)-nordeoxyelephantopin and its analogues. Target identification experiments revealed eleven previously unreported selective targets of deoxyelephantopin that could be responsible for its cytotoxicity. Since it was already known that deoxyelephantopin is a partial PPAR γ - agonist, the authors focused on their interactions. The paper is interesting to read, the total synthesis of great value and the target identification of this potent natural product is very important, however some major issues need to be addressed in order to be suitable for publication.

1. All new compounds are characterized by ¹H and ¹³C-NMR spectra. The scans of NMR spectra are also provided. Many of these spectra indicate sufficient amounts of impurities. Some of these impurities can be identified as solvents (such as ethyl acetate), the others are unknown. I do understand that in many cases only a minute amounts of compounds were synthesized and it is not that easy to obtain them in a completely pure form but at least the most important ones (nordeoxyelephantopin, ent- nordeoxyelephantopin or probes 24 a-c used for the labeling) should be isolated in a completely pure form (purifying them, for example, by preparative HPLC). For many intermediates no mass-spectra are provided at all (Standard characterization requires ¹H+¹³C- NMR + mass spectra). These mass spectra should be measured. For the key substances the authors measured low resolution mass-spectra. That is also not sufficient. In order to prove the molecular formula and purity of the substance it is necessary to have either elemental analysis data or a combination of high resolution mass spectra and LC. The authors should provide them here.

2. The authors aim to identify the targets of elephantopins. It is not clear why gamma-lactone probes 23 and 24 were used for competitive profiling and not the original natural product substituted with an alkyne. It cannot be excluded that in addition to the competed proteins elephantopins bind to other proteins as well. A rationale for this approach should be provided.

3. The authors identify 11 putative binders which were not reported as targets of elephantopins before. While this is an interesting observation which would merit some closer target validation, the authors focus on PPAR γ as a target although this was not detected. In the manuscript it is speculated that this is due to its low abundance in MCF7. Validation was then indirectly carried out by spiking of recombinant PPAR γ into MCF7 lysates. This is not convincing, as it is highly artificial. I recommend to repeat these experiments with a cell line that expresses higher amounts of PPAR γ . Alternatively, it would be interesting to validate some of the other targets. At the moment the reader gets the impression that all the target finding efforts are irrelevant since a target that was not detected is more important and thus validated in much detail.

4. The authors identify the PPAR γ binding site and state that deoxyelephantopin addresses a zinc finger through a Michael addition. This is only indirectly shown as the probe 23c was used to identify the site. It is recommended to repeat these experiments with the natural product to confirm this binding.

5. "preliminary experiments on the reactivity..." no data provided. Must be included

6. "probe 24c that showed highest cytotoxicity among the entire series" there is no statistics provided in Figure 2A to make such a statement. Are differences significant?

7. "only few bands were selectively competed by twofold excess of the natural product" I have a hard time to see these few bands that are selectively competed. It looks like everything gets lighter. Remove gel to SI and rephrase sentence.

Reviewer #2 (Remarks to the Author):

Winssinger and coworkers report the synthesis of a series of analogues of deoxyelephantopin, a naturally occurring a sesquiterpene lactone with anticancer activities. The main challenge for the synthesis was the construction of the strained ten-membered ring. The authors employed a

desymmetrizing RCM reaction to prepare a racemic butenolide intermediate. A Zn-mediated allylation was responsible for the coupling of the two fragments and establishing three stereogenic centers. Another RCM reaction was then used to assemble the strained medium-sized ring. The synthetic route is short and efficient. The authors cleverly chose nordeoxyelephantopin as targets, which reduced the synthetic difficulty (the second RCM reaction) yet remained the biological activity. The asymmetric version of the synthesis was achieved by using Trost's enantioselective allylic substitution. An interesting observation is that swapping the order of Ni-catalyzed CO insertion and medium-sized ring formation resulted in different geometry of C4=C5 bond. A photoinduced cyclization was described, which gave an unprecedented bridged lactone system.

The authors demonstrated that some of the synthetic analogues displayed potent cytotoxicity against cancer cell lines. They then exploited a Cravatt-type strategy to perform a competitive proteomic profiling, taking advantage of the electrophilic motif of the molecules. The identified targets are worthy of further studies with these small molecules. The attention was finally directed to PPAR γ , and the exact binding site with the small molecule probe was identified by using mass spectrometry technique.

This manuscript describes a full and inspiring story of chemical synthesis and biological studies of sesquiterpene lactones. This reviewer supports its publication in Nat. Commun.

Reviewer #3 (Remarks to the Author):

Summary

Lagoutte et al. developed the synthesis of deoxyelephantopin analogues based on the Barbier reaction and ring closing metathesis. Using an alkyne containing deoxyelephantopin derivative the authors identified 11 proteins in a quantitative proteomics-based competition binding assay. These 11 proteins include cancer-related proteins like CTTN, CSTB, and CBS. Additionally the authors spiked the recombinant transcription factor PPAR γ - a known target of deoxyelephantopin - into cell lysates and successfully enriched the protein. Lagoutte et al. were able to obtain the kinetic values K_i and k_{inact} for deoxyelephantopin and confirmed that the sesquiterpene lactone irreversibly binds to PPAR γ via the zinc-bound Cys190 in a zinc finger motif of the protein.

The manuscript is mostly well written and the chemistry findings will be of interest to a subset of readers of Nature Communications, and the chemistry presented is well-executed, apart from some missing spectral data. However, there are significant scientific and technical issues in the execution of probe design and application that preclude acceptance without further experiments to ensure that the biological data are robust, and that the conclusions drawn from them are valid.

Key issues:

The authors developed a straightforward synthesis strategy to obtain their target molecules. The provided analytical data is conclusive but partially incomplete. HRMS spectra of most of the intermediary products are entirely missing and no alternative characterization methods were used. In general, every molecule should be fully characterized by NMR (1H and ^{13}C) and, additionally, HRMS. Another incomplete characterization concerns the stereochemistry of the molecules. Although a NOESY spectrum of nordeoxyelephantopin was recorded (the supporting information contain a very brief schematic figure that highlights the NOESY correlations), the spectrum itself is not included. Additionally the authors used NMR coupling constants of nordeoxyelephantopin and the natural deoxyelephantopin to show that both compounds exhibit very similar conformations. At least NOESY spectra of the final compounds are necessary to prove that they exhibit the right conformation. A comparison with the NOESY spectra of natural deoxyelephantopin would prove the conformational analogy.

A brief comment is made on relative reactivity of the three conjugate systems, but no experimental evidence is provided; this should be presented (in SI) and discussed in the context of conformational strain in the molecule. The probes that underpin the biological analysis are based on this unsupported statement, and it seems very likely that the targets engaged by the probes will be different due to the lack of conformational constraint, resulting in loss of potential targets in competitive profiling experiments.

Critically, all the target profiling experiments, from gels to proteomics, have been performed in a non-physiological system - cell lysates - as opposed to in intact cells. This fact is somewhat hidden in the MS, since Fig. 3B seems to imply that profiling is done in cells, e.g. in the legend: "Experimental workflow for the competitive proteomic profiling of DEP targets in SILAC-labeled MCF-7 cells. Proteomic targets of DEP (>70% competition) in MCF-7 cells identified in a SILAC experiment (n = 2)." This misleading legend should be phrased as in cell lysates, not in cells.

The decision to profile in lysates is puzzling, particularly given the expertise of the Adebekian lab in indirect cellular target ID by reactive Cys profiling (as pioneered in the Cravatt lab), and significantly undermines the biological relevance of the data. For example, it is highly probable that the probe reacts rapidly with glutathione on entering the cell, resulting in redox changes and target profiles that cannot be recapitulated by cell-free (lysate) profiling. The natural product should be used as the probe scaffold, thanks to the access provided by the elaborate synthetic strategy developed here, and the targets should be profiled directly in cells to provide data that could be physiologically relevant. This could be done either directly using a close analogue of DEP with an alkyne attached, or indirectly using target engagement in cells by DEP, followed by cysteine profiling using standard tools for Cys profiling in lysates (e.g. Cravatt's IsoTOP-ABPP), a particularly area of expertise of one of the authors. Cell-based (as opposed to purely lysate-based) data would be much more robust, will provide more dependable modification site IDs, and crucially will provide much more physiologically relevant data.

Cell cytotoxicity of natural deoxyelephantopin and the corresponding analogues were only tested at a concentration of 1 μ M, in MCF-7 cells. No rationale is provided for choosing this cell line, and given the very small panel of compounds tested there is no obvious reason why an IC₅₀ (dose-response) should not be determined to facilitate comparison of their potencies, and across a small panel of cancer cells. No detail is provided on the assay used to determine viability; this is key, since many assays simply measure metabolism or DNA synthesis, not whether a cell remains viable, and underlie a large amount of the well-known poor reproducibility of cell cytotox assay data. In Fig S1 it is difficult to assess changes in cell morphology; a picture of greater magnification would be better, but the only visible change is a decrease number of cells in the DEP-treated sample - this could equally have been a result of e.g. uneven seeding of cells. Confident claims of cell death by apoptosis require a minimum of two orthogonal lines of evidence since it is possible to see caspase activation in the presence of cell death by other mechanisms. For example flow cytometry analysis using Annexin V and propidium iodide is sufficient to confidently assign a population of cells to apoptosis, in the presence of caspase activation.

Given the lack of cell-based profiling, the important question of whether the compounds actually enter the cell has also not been addressed; this could be demonstrated by proving site-specific modification by DEP at a target protein (e.g. PPAR γ) following exposure in cells. Otherwise, it remains plausible that DEP acts through surface receptor interactions or non-specific plasma membrane interactions, causing changes in redox state that are picked up by the probe following lysis.

The findings of the SILAC based identification of deoxyelephantopin targets are potentially interesting, but require additional validation - and critically, need to be linked to relevant cell-based target engagement (see comments above). An arbitrary cut-off (>70% competition) has been chosen without justification, and there has been very little analysis of the dataset generated. This should be performed using standard tools for pathway analysis (DAVID, Cytoscape), and

compared with whole proteome changes previously reported (ref 26) and DEP's several reported bioactivities (e.g. NFκB suppression). Valid target pathways should be expected to map onto aspects of the hit set reported in the present MS.

It is clearly unlikely that PPARγ has any relevance to the cytotoxicity observed given weak binding and slow inactivation by DEP, and the fact that weakly cytotoxic compounds (19a) bind it more readily than DEP itself. Furthermore, ref 27 claims DEP is a partial PPARγ agonist, an observation at odds with the present data. It would be sensible to determine PPARγ activity through a bioassay in cells using the pure material synthetic material in the present MS; preparations used in the literature may contain minor bioactive contaminants, and it would be important to demonstrate whether authentic DEP is agonistic or antagonistic. Alternatively, and more likely, this is an example of purely lysate-based experiments providing physiologically irrelevant data - it is entirely plausible that DEP never engages PPARγ directly in the cell, due to inactivation by glutathione and/or target compartmentalization, for example. Given these issues, there is no strong evidence in the present MS as to whether any of the targets identified are relevant to the biology of DEP, fundamentally undermining the target data presented. It is critical to provide evidence for target engagement in cells, not merely in lysates.

The MS2 spectrum in Fig 5A requires better annotation for confident assignment, since there are intense peaks that remain unassigned; fragmentation of the site chain modification should be considered, only one b ion is annotated, and important b3 and y5 ions are missing. Regarding docking experiments, pull-down assays using probe 19a and point mutants of PPARγ (e.g. Asp174Ala and Cys176Ala) could be used to prove these findings. If (e.g. in the case of Asp174Ala) PPARγ would still be detected by Western Blot analysis of the pull-down fraction, a PMF analysis of the protein should be performed to prove whether or not the binding site is still the same.

Minor issues:

There remain a good number of typos in the MS (see below), but otherwise the clarity of the text is satisfactory.

Substantial revision of the figure legends is required; it should be possible to understand the content of the figures without detailed reading of the SI methods, and there are many key details not mentioned.

The changes in the band pattern in the gel-based competitive proteomic profiling assay are faint. Taking this into consideration, the arrangement and quality of Figure 3 is rather poor and therefore not appropriate. There are large empty areas e.g. due to the alignment of the caption of the gel profiles. The authors should rearrange the Figure and significantly increase the size of the gel images.

A figure (at least in the SI) presenting the structures of all final probes applied in cells would further improve the readability of the manuscript, particularly for biologists, and Deoxyelephantopin and ent-nordeoxyelephantopin should be numbered in the text.

Minor edits/typos:

Herbal extracts containing [...]

[...] functionalities that can engage in covalent interactions [...]

[...] groups such as α-methylene-γ-butyrolactone, α,β-unsaturated reactive ester chain, and epoxides

Helenalin is broadly used as an anti-inflammatory drug in the form [...]

[...]without affecting protein geranylgeranylation.

[...]transcriptional regulators and inhibiting the virulence of *Staphylococcus aureus*. (*S. aureus* in italics)

[...] Baylis-Hillman reaction with formaldehyde provided the cyclization [...]

[...]on its propensity to isomerize[...]

[...] was pursued to access analogues wherein [...]

[...] As an alternative, substrate 13 was converted [...]

Enantiomerically enriched (R)-6 was [...]

We determined that the cell death is caused by caspase-mediated apoptosis by treating MCF-7 cells with 20 μ M[...] - of which compounds?

Reviewers' comments:

Reviewer #1 (Remarks to the Author):

In the manuscript entitled 'Divergent synthesis of deoxyelephantopins and investigations into its covalent interactome' by Winssinger et al. authors describe synthesis and biological activity of (+/-)-nordeoxyelephantopin and its analogues. Target identification experiments revealed eleven previously unreported selective targets of deoxyelephantopin that could be responsible for its cytotoxicity. Since it was already known that deoxyelephantopin is a partial PPAR γ - agonist, the authors focused on their interactions. The paper is interesting to read, the total synthesis of great value and the target identification of this potent natural product is very important, however some major issues need to be addressed in order to be suitable for publication.

1. All new compounds are characterized by ^1H and ^{13}C -NMR spectra. The scans of NMR spectra are also provided. Many of these spectra indicate sufficient amounts of impurities. Some of these impurities can be identified as solvents (such as ethyl acetate), the others are unknown. I do understand that in many cases only a minute amounts of compounds were synthesized and it is not that easy to obtain them in a completely pure form but at least the most important ones (nordeoxyelephantopin, ent-nordeoxyelephantopin or probes 24 a-c used for the labeling) should be isolated in a completely pure form (purifying them, for example, by preparative HPLC).

For many intermediates no mass-spectra are provided at all (Standard characterization requires $^1\text{H}+^{13}\text{C}$

C- NMR + mass spectra). These mass spectra should be measured. For the key substances the authors measured low resolution mass-spectra. That is also not sufficient. In order to prove the molecular formula and purity of the substance it is necessary to have either elemental analysis data or a combination of high resolution mass spectra and LC. The authors should provide them here.

-> NMR spectra of final compounds have been re-measured on purified material. HRMS for all requested compounds have been measured and added to the SI (16 compounds, final products and key precursors: R-10; S-10; nordeoxyelephantopin; *ent*-nordeoxyelephantopin; 11; 12; 18; 19a; 19b; 20a; 20b; 23a; 23b; 23c; 24a; 24b; 24c; 24c-Cy3; 26; 28)

2. The authors aim to identify the targets of elephantopins. It is not clear why gamma-lactone probes 23 and 24 were used for competitive profiling and not the original natural product substituted with an alkyne. It cannot be excluded that in addition to the competed proteins elephantopins bind to other proteins as well. A rationale for this approach should be provided.

-> Appending an alkyne on elephantopins may lead to loss of binding and multiple permutations would need to be investigated to rule out that a given permutation was not capturing only a subset of the targets. A more flexible probe that recapitulates the biological activity of the rigid natural product was deemed a safer alternative using competition experiments. Cellular cytotoxicity assays pointed to 24c as a suitable candidate to this end. During our gel-based competition studies, we discovered that probe 24c and deoxyelephantopin exhibit very similar proteome reactivity *in situ* (Figure 3B). Briefly, live MCF7 cells were treated with 24c or deoxyelephantopin, lysed, and then treated with the fluorescent probe 24c-Cy3. Probe 24c-Cy3 labeled a variety of different proteins and pretreatment with both compounds resulted in a largely overlapping competition pattern. Hence, decision was made to employ 24c as proteomic probe in subsequent experiments. The following sentence was added to the text to clarify this point: "While the greater conformational flexibility of probe **24c** could potentially lead to more promiscuous target engagement relatively to deoxyelephantopin, it was favored based on the fact that an alkyne moiety appended to the rigid scaffold of deoxyelephantopin may hinder some of its interactions."

3. The authors identify 11 putative binders which were not reported as targets of elephantopins before. While this is an interesting observation which would merit some closer target validation, the authors focus on PPAR γ as a target although this was not detected. In the manuscript it is speculated that this is due to its low abundance in MCF7. Validation was then indirectly carried out by spiking of recombinant PPAR γ into MCF7 lysates. This is not convincing, as it is highly artificial. I recommend to repeat these experiments with a cell line that expresses higher amounts of PPAR γ . Alternatively, it would be interesting to validate some of the other targets. At the moment the reader gets the impression that all the target finding efforts are irrelevant since a target that was not detected is more important and thus validated in much detail.

-> We thank the referee for this constructive criticism. Experiments have now been added with a cell line that expresses higher amount of PPAR γ (Caco-2). These new experiments provide compelling evidence that deoxyelephantopin binds and covalently engages endogenous PPAR γ directly in live Caco-2 cells. Firstly, we show by LC-MS/MS that endogenous PPAR γ in Caco-2 cells can be enriched by 24c. Secondly, PPAR γ enrichment is >95% abolished when living cells are pretreated with 20 μ M

deoxyelephantopin (Figure 4A). Moreover, we expressed PPAR γ in HeLa cells and treated these cells *in situ* with 20 μ M DEP followed by *in vitro* treatment 24c-Cy3 which again showed efficient competition of PPAR γ labeling (Figure S9). Finally, comparison of the expression levels of four reported PPAR γ -regulated proteins after 24 h treatment of Caco-2 cells with deoxyelephantopin vs known PPAR γ antagonist/agonist reveals that deoxyelephantopin acts as an antagonist of PPAR γ *in situ* (Figure 4B, Table S4). We decided to focus on PPAR γ due to the high therapeutic relevance of this protein among all identified DEP targets. This is now clearly stated in the revised manuscript.

4. The authors identify the PPAR γ binding site and state that deoxyelephantopin addresses a zinc finger through a Michael addition. This is only indirectly shown as the probe 23c was used to identify the site. It is recommended to repeat these experiments with the natural product to confirm this binding.

-> Unfortunately, it was not possible to directly identify deoxyelephantopin-PPAR γ adduct via MS/MS due to the complex fragmentation pattern of this natural product. However, the IAA-PPAR γ adduct on the tryptic peptide with Cys190 was not detectable when the protein was first treated with DEP followed by iodoacetamide (Table S5) concurring the fact that this cysteine is covalently engaged to deoxyelephantopin. The fact that a covalent adduct can be observed by MS with probe 23c coupled to the fact that deoxyelephantopin competes with PPAR γ labeling of 23c and IAA point to the fact that deoxyelephantopin reacts with Cys190. Moreover, because our docking model suggests key interactions between the structurally close derivative 19a and Asp174 and Cys176, we mutated these two residues and observed a shift in the IC₅₀ value of 19a binding from 19 μ M (wt) to 33 μ M (C176A) and 58 μ M (D174A). This result further confirms the previously suggested binding site for deoxyelephantopin and its derivatives.

5. "preliminary experiments on the reactivity..." no data provided. Must be included

-> The NMR and LCMS data have now been added to the SI. The following sentence has been added to the text for clarity: "reaction with 5 equivalent of glutathione led to a single addition product onto the γ -butyrolactone, see SI for full characterization"

6. "probe 24c that showed highest cytotoxicity among the entire series" there is no statistics provided in Figure 2A to make such a statement. Are differences significant?

-> The cytotoxicity has now been repeated in four different cell lines (MCF7, Cacao-2, HeLa, MDA-MB-231) and the IC₅₀ values determined for deoxyelephantopin, 19a, and the alkyne probes 23a-c and 24a-c with statistical analysis (Figure 2A). Probe 24c consistently exerted the highest toxicity among all alkyne probes in the four cell lines. Moreover, two-sided Student's t-test analysis revealed that the difference in cytotoxicity between 24c and the second most cytotoxic probe(s) is statistically significant.

7. "only few bands were selectively competed by twofold excess of the natural product" I have a hard time to see these few bands that are selectively competed. It looks like everything gets lighter. Remove gel to SI and rephrase sentence.

-> We present a new gel that shows concentration-dependent competition of 24c-Cy3 labeling with DEP (Figure 3A). Most importantly, deoxyelephantopin treatment was now performed in live MCF7 cells rather than lysates and clear competition of several distinct bands can be seen.

Reviewer #2 (Remarks to the Author):

Winssinger and coworkers report the synthesis of a series of analogues of deoxyelephantopin, a naturally occurring sesquiterpene lactone with anticancer activities. The main challenge for the synthesis was the construction of the strained ten-membered ring. The authors employed a desymmetrizing RCM reaction to prepare a racemic butenolide intermediate. A Zn-mediated allylation was responsible for the coupling of the two fragments and establishing three stereogenic centers. Another RCM reaction was then used to assemble the strained medium-sized ring. The synthetic route is short and efficient. The authors cleverly chose nordeoxyelephantopin as targets, which reduced the synthetic difficulty (the second RCM reaction) yet remained the biological activity. The asymmetric version of the synthesis was achieved by using Trost's enantioselective allylic substitution. An interesting observation is that swapping the order of Ni-catalyzed CO insertion and medium-sized ring formation resulted in different geometry of C4=C5 bond. A photoinduced cyclization was described, which gave an unprecedented bridged lactone system.

The authors demonstrated that some of the synthetic analogues displayed potent cytotoxicity against cancer cell lines. They then exploited a Cravatt-type strategy to perform a competitive proteomic profiling, taking advantage of the electrophilic motif of the molecules. The identified targets are worthy of further studies with these small molecules. The attention was finally directed to PPAR γ , and the exact binding site with the small molecule probe was identified by using mass spectrometry technique.

This manuscript describes a full and inspiring story of chemical synthesis and biological studies of sesquiterpene lactones. This reviewer supports its publication in Nat. Commun.

Reviewer #3 (Remarks to the Author):

Summary

Lagoutte et al. developed the synthesis of deoxyelephantopin analogues based on the Barbier reaction and ring closing metathesis. Using an alkyne containing deoxyelephantopin derivative the authors identified 11 proteins in a quantitative proteomics-based competition binding assay. These 11 proteins include cancer-related proteins like CTTN, CSTB, and CBS. Additionally the authors spiked the recombinant transcription factor PPAR γ - a known target of deoxyelephantopin - into cell lysates and successfully enriched the protein. Lagoutte et al. were able to obtain the kinetic values K_i and k_{inact} for deoxyelephantopin and confirmed that the sesquiterpene lactone irreversibly binds to PPAR γ via the zinc-bound Cys190 in a zinc finger motif of the protein.

The manuscript is mostly well written and the chemistry findings will be of interest to a subset of readers of Nature Communications, and the chemistry presented is well-executed, apart from some missing spectral data. However, there are significant scientific and technical issues in the execution of probe design and application that preclude acceptance without further experiments to ensure that the biological data are robust, and that the conclusions drawn from them are valid.

Key issues:

The authors developed a straightforward synthesis strategy to obtain their target molecules. The provided analytical data is conclusive but partially incomplete. HRMS spectra of most of the intermediary products are entirely missing and no alternative characterization methods were used. In general, every molecule should be fully characterized by NMR (^1H and ^{13}C) and, additionally, HRMS. Another incomplete characterization concerns the stereochemistry of the molecules. Although a NOESY spectrum of nordeoxyelephantopin was recorded (the supporting information contain a very brief schematic figure that highlights the NOESY correlations), the spectrum itself is not included. Additionally the authors used NMR coupling constants of nordeoxyelephantopin and the natural deoxyelephantopin to show that both compounds exhibit very similar conformations. At least NOESY spectra of the final compounds are necessary to prove that they exhibit the right conformation. A comparison with the NOESY spectra of natural deoxyelephantopin would prove the conformational analogy.

-> As requested by reviewer 1, HRMS data has been added for all final compounds. The stereochemical purity of compound 6 was measured by chiral GC (92:8 e.r.; spectra shown in the SI). The NOESY spectra of the nordeoxyelephantopin and deoxyelephantopin have been added and compared. The comparison of the coupling constant had been favored as a measure of conformational similarity between the two compounds based on the fact that the 10-membered ring is very rigid and the dihedral angle at each carbon is clearly reflected in the coupling constants. This analysis is now complemented by the NOESY comparison that indeed show comparable proximity for protons on either face of the 10-membered ring. In addition NOESY and COSY spectra of compounds 19a, 19b, 20a and 20b are included.

A brief comment is made on relative reactivity of the three conjugate systems, but no experimental evidence is provided; this should be presented (in SI) and discussed in the context of conformational strain in the molecule. The probes that underpin the biological analysis are based on this unsupported statement, and it seems very likely that the targets engaged by the probes will be different due to the lack of conformational constraint, resulting in loss of potential targets in competitive profiling experiments.

-> As requested by reviewer 1, the NMR and LCMS data from the reaction of nordeoxyelephantopin with glutathione have now been added to the SI. The data clearly shows that the γ -butyrolactone is the most reactive Michael acceptor.

Critically, all the target profiling experiments, from gels to proteomics, have been performed in a non-physiological system - cell lysates - as opposed to in intact cells. This fact is somewhat hidden in the MS, since Fig. 3B seems to imply that profiling is done in cells, e.g. in the legend: "Experimental workflow for the competitive proteomic profiling of DEP targets in SILAC-labeled MCF-7 cells. Proteomic targets of DEP (>70% competition) in MCF-7 cells identified in a SILAC experiment (n = 2)." This misleading legend should be phrased as in cell lysates, not in cells.

The decision to profile in lysates is puzzling, particularly given the expertise of the Adebekian lab in indirect cellular target ID by reactive Cys profiling (as pioneered in the Cravatt lab), and significantly undermines the biological relevance of the data. For example, it is highly probable that the probe reacts rapidly with glutathione on entering the cell, resulting in redox changes and target profiles that cannot be recapitulated by cell-free (lysate) profiling. The natural product should be used as the probe scaffold, thanks to the access provided by the elaborate synthetic strategy developed here, and the targets should be profiled directly in cells to provide data that could be physiologically relevant. This could be done either directly using a close analogue of DEP with an alkyne attached, or indirectly using target engagement in cells by DEP, followed by cysteine profiling using standard tools for Cys profiling in lysates (e.g. Cravatt's IsoTOP-ABPP), a particularly area of expertise of one of the authors. Cell-based (as opposed to purely lysate-based) data would be much more robust, will provide more dependable modification site IDs, and crucially will provide much more physiologically relevant data.

-> We agree with the Reviewer that the proteomics experiments in lysates are physiologically less relevant. Therefore, we performed a series of additional experiments to investigate the activity/target engagement of deoxyelephantopin in living cells. Firstly, we include a gel showing concentration-dependent competition of 24c-Cy3 labeling with deoxyelephantopin. Deoxyelephantopin treatment was this time performed directly in living MCF7 cells and clear competition of several distinct bands can be seen (Figure 3A). Secondly, we have now performed a targeted competitive proteomics profiling experiment directly in intact MCF7 cells and confirmed that all 11 previously mentioned targets are also >70% competed by deoxyelephantopin *in situ* (Figure 3C, Table S2).

Moreover, we have now performed a series of new experiments to provide compelling evidence that deoxyelephantopin (and derivatives) engage and bind endogenous PPAR γ directly in intact Caco-2 cells (a cell line that is known to express higher levels of PPAR γ). We show by LC-MS/MS that endogenous PPAR γ in Caco-2 cells can be enriched by 24c and that PPAR γ enrichment is >95% abolished when living cells are pretreated with 20 μ M DEP (Figure 4A). Furthermore, we expressed PPAR γ in HeLa cells and treated these cells *in situ* with 20 μ M DEP followed by *in vitro* treatment 24c-Cy3; concurring the efficient competition of PPAR γ labeling (Figure S9). Finally, comparison of the expression levels of five known PPAR γ -regulated proteins after 24 h treatment of Caco-2 cells with deoxyelephantopin clearly establish that the deoxyelephantopin acts as an antagonist of PPAR γ *in situ* (Figure 4B, Table S4). While any Michael acceptor will ultimately be inactivated by cellular thiols, the data shows that covalent target engagement is faster than inactivation of the product by glutathione or other cellular thiols.

The point regarding the use of an alkyne-tagged version of the natural product as a probe has also been raised by reviewer 1 (point 2). As argued above, without a priori knowledge of the binding mode of deoxyelephantopin, such probe cannot be designed without potentially compromising its interactions with a subset of its targets. We thus favored competition experiments with simplified analog (24c) wherein the alkyne is unlikely to preclude target binding due to steric clashes.

Cell cytotoxicity of natural deoxyelephantopin and the corresponding analogues were only tested at a concentration of 1 μ M, in MCF-7 cells. No rationale is provided for choosing this cell line, and given the very small panel of compounds tested there is no obvious reason why an IC₅₀ (dose-response) should

not be determined to facilitate comparison of their potencies, and across a small panel of cancer cells. No detail is provided on the assay used to determine viability; this is key, since many assays simply measure metabolism or DNA synthesis, not whether a cell remains viable, and underlie a large amount of the well-known poor reproducibility of cell cytotoxic assay data. In Fig S1 it is difficult to assess changes in cell morphology; a picture of greater magnification would be better, but the only visible change is a decrease number of cells in the DEP-treated sample - this could equally have been a result of e.g. uneven seeding of cells. Confident claims of cell death by apoptosis require a minimum of two orthogonal lines of evidence since it is possible to see caspase activation in the presence of cell death by other mechanisms. For example flow cytometry analysis using Annexin V and propidium iodide is sufficient to confidently assign a population of cells to apoptosis, in the presence of caspase activation.

-> We have now measured the IC_{50} values for deoxyelephantopin, 19a, and the alkyne probes 23a-c and 24a-c in four different cancer cell lines (MCF7, Caco-2, HeLa, MDA-MB-231; Figure 2A). Cell viability was determined by nuclei count and the experimental details are provided in the SI. Fig. S2 is now shown in greater magnification to emphasize morphological changes caused by deoxyelephantopin treatment. Evidence for cell death via apoptosis caused by deoxyelephantopin treatment is now shown in MCF7 cells using two microscopy-based orthogonal methods, Annexin V staining and Caspase activity staining (Figure 2B). Moreover, we performed the suggested propidium iodide experiment that excluded the possibility of cell death through necrosis (Figure S4).

Given the lack of cell-based profiling, the important question of whether the compounds actually enter the cell has also not been addressed; this could be demonstrated by proving site-specific modification by DEP at a target protein (e.g. PPAR γ) following exposure in cells. Otherwise, it remains plausible that DEP acts through surface receptor interactions or non-specific plasma membrane interactions, causing changes in redox state that are picked up by the probe following lysis.

-> This is an excellent point. We now profiled Caco-2 cells showing that probe 24c efficiently pulls down PPAR γ and that this pull-down is inhibited by deoxyelephantopin administered to intact cells. The same experiment was performed in intact HeLa expressing hPPAR γ .

The findings of the SILAC based identification of deoxyelephantopin targets are potentially interesting, but require additional validation - and critically, need to be linked to relevant cell-based target engagement (see comments above). An arbitrary cut-off (>70% competition) has been chosen without justification, and there has been very little analysis of the dataset generated. This should be performed using standard tools for pathway analysis (DAVID, Cytoscape), and compared with whole proteome changes previously reported (ref 26) and DEP's several reported bioactivities (e.g. NF κ B suppression). Valid target pathways should be expected to map onto aspects of the hit set reported in the present MS. It is clearly unlikely that PPAR γ has any relevance to the cytotoxicity observed given weak binding and slow inactivation by DEP, and the fact that weakly cytotoxic compounds (19a) bind it more readily than DEP itself. Furthermore, ref 27 claims DEP is a partial PPAR γ agonist, an observation at odds with the present data. It would be sensible to determine PPAR γ activity through a bioassay in cells using the pure material synthetic material in the present MS; preparations used in the literature may

contain minor bioactive contaminants, and it would be important to demonstrate whether authentic DEP is agonistic or antagonistic. Alternatively, and more likely, this is an example of purely lysate-based experiments providing physiologically irrelevant data - it is entirely plausible that DEP never engages PPAR γ directly in the cell, due to inactivation by glutathione and/or target compartmentalization, for example. Given these issues, there is no strong evidence in the present MS as to whether any of the targets identified are relevant to the biology of DEP, fundamentally undermining the target data presented. It is critical to provide evidence for target engagement in cells, not merely in lysates.

-> We have now performed a comparative global proteomics experiment to map proteome-wide PPAR γ -dependent changes in protein levels in Caco-2 cells following 20 μ M DEP treatment vs treatment with known PPAR γ agonists (10 μ M rosiglitazone and 50 μ M antagonist T0070907). We identified 25 upregulated and 18 downregulated proteins following treatment with both DEP and T0070907 and these proteins were not or differently affected by rosiglitazone treatment, indicating that DEP acts in cells as an antagonist (partial agonist) of PPAR γ (Table S4). These 43 dysregulated proteins were not reported in ref. 26. This is not surprising, because the proteomics experiments in ref. 26 were performed in a completely different cell line that may not express sufficient amounts of PPAR γ . The list of 43 dysregulated proteins also includes four reported PPAR γ -dependent proteins RBP4, GSTA2, SLC26A3 and LIPA (Figure 4B). Most strikingly, RBP4, a well-studied contributor to insulin resistance in obesity and type 2 diabetes and a major therapeutic target, was significantly more downregulated following 24 h treatment with DEP vs. T0070907, suggesting that derivatives of DEP or a derivative may potentially serve as antidiabetic drugs. Interestingly, T0070907 was also reported to suppress breast cancer cell proliferation in PPAR γ -dependent manner, meaning that at least some part of the cytotoxic activity of deoxyelephantopin may indeed originate from antagonizing this receptor. Finally, we performed GOrilla pathway analysis of the 43 dysregulated proteins (Figure S10).

A cut-off of 70% was chosen to present the most efficiently competed *in vitro* DEP targets in Figure S7. However, we also present the full list of DEP targets in Table S1.

The MS2 spectrum in Fig 5A requires better annotation for confident assignment, since there are intense peaks that remain unassigned; fragmentation of the site chain modification should be considered, only one b ion is annotated, and important b3 and y5 ions are missing. Regarding docking experiments, pull-down assays using probe 19a and point mutants of PPAR γ (e.g. Asp174Ala and Cys176Ala) could be used to prove these findings. If (e.g. in the case of Asp174Ala) PPAR γ would still be detected by Western Blot analysis of the pull-down fraction, a PMF analysis of the protein should be performed to prove whether or not the binding site is still the same.

-> The MS2 spectrum (23c-PPAR γ adduct) in has now been interrogated manually to identify additional diagnostic ions and unambiguously confirm the exact modification site on Cys190 (Fig. S14):

- IAA competition experiments indicate that the adduct is very likely formed on a cysteine residue. Since this peptide contains two cysteines, it is important to verify that Cys193 is not modified. This is unambiguously confirmed by ions y_4 , y_3 , y_2 and a_5 .

Moreover, a less likely possibility of a modification on Lys189 as another nucleophilic amino acid is excluded by ion b_2 .

In addition, because our docking model suggests key interactions between the structurally close DEP derivative 19a and Asp174 and Cys176 on PPARG, we mutated these two residues and performed gel-based competitive experiments to determine relative binding affinities for 19a. We observed a shift in the IC50 value from 19 μ M (wt PPARG) to 33 μ M (C176A) and 58 μ M (D174A) (Figures 5B, S16). This result further confirms the suggested binding site for 19a.

Minor issues:

There remain a good number of typos in the MS (see below), but otherwise the clarity of the text is satisfactory.

Substantial revision of the figure legends is required; it should be possible to understand the content of the figures without detailed reading of the SI methods, and there are many key details not mentioned.

-> Key experimental details have been added to the figure legends.

The changes in the band pattern in the gel-based competitive proteomic profiling assay are faint. Taking this into consideration, the arrangement and quality of Figure 3 is rather poor and therefore not appropriate. There are large empty areas e.g. due to the alignment of the caption of the gel profiles. The authors should rearrange the Figure and significantly increase the size of the gel images.

-> The gel in Figure 3 has been replaced by a new gel that shows concentration-dependent competition of 24c-Cy3 labeling with DEP (Figure 3A). Most importantly, DEP treatment was this time performed in intact MCF7 cells and clear competition of several distinct bands can be seen.

A figure (at least in the SI) presenting the structures of all final probes applied in cells would further improve the readability of the manuscript, particularly for biologists, and Deoxyelephantopin and ent-nordeoxyelephantopin should be numbered in the text.

-> A summary figure with final compounds has been added to the SI as suggested. We have opted to use proper name rather than a number for the natural product because we find the prose easier to understand.

Minor edits/typos:

Herbal extracts containing [...]

[...] functionalities that can engage in covalent interactions [...]

[...] groups such as α -methylene- γ -butyrolactone, α,β -unsaturated reactive ester chain, and epoxides

Helenalin is broadly used as an anti-inflammatory drug in the form [...]

[...]without affecting protein geranylgeranylation.

[...]transcriptional regulators and inhibiting the virulence of *Staphylococcus aureus*. (*S. aureus* in italics)

[...] Baylis-Hillman reaction with formaldehyde provided the cyclization [...]

[...]on its propensity to isomerize[...]

[...] was pursued to access analogues wherein [...]

[...] As an alternative, substrate 13 was converted [...]

Enantiomerically enriched (R)-6 was [...]

We determined that the cell death is caused by caspase-mediated apoptosis by treating MCF-7 cells with 20 μ M[...] - of which compounds?

-> Typos have been corrected and the manuscript has been further proofread.

Reviewer #1 (Remarks to the Author):

The new manuscript addresses all my previous concerns and is now suitable for publication.

Reviewer #3 (Remarks to the Author):

The authors have satisfactorily addressed the comments of the reviewers.